# Can Endocrine Dysfunction Be Reliably Tested in Aged Horses That Are Experiencing Pain?

**DOI:** 10.3390/ani10081426

**Published:** 2020-08-14

**Authors:** Heidrun Gehlen, Nina Jaburg, Roswitha Merle, Judith Winter

**Affiliations:** 1Equine Clinic, Freie University of Berlin, 14163 Berlin, Germany; nina_jaburg@yahoo.de (N.J.); Judith.Winter@synlab.com (J.W.); 2Veterinary Department, Institute of Veterinary Epidemiology, Freie University Berlin, 14163 Berlin, Germany; Roswitha.Merle@fu-berlin.de

**Keywords:** pain, pain scoring, ACTH, cortisol, PPID, horse, endocrine disease

## Abstract

**Simple Summary:**

Pituitary pars intermedia dysfunction (PPID) is an endocrine (secreting internally) disease of aged horses and ponies. An enlargement (hyperplasia) of the pars intermedia of the pituitary gland leads to an increased secretion of hormones, including adrenocorticotropic hormone (ACTH). The main tests for a diagnosis of PPID are the measurement of basal ACTH and the thyrotropin-releasing hormone (TRH) stimulation test, where TRH stimulates the secretion of ACTH. Since pain can also lead to elevated concentrations of ACTH, it is unclear whether horses with pain can be tested for PPID correctly. The aim of the present study was to find out whether pain caused a marked increase of ACTH can lead to a false positive result in the diagnosis of PPID. Therefore, we examined fifteen horses treated for different pain conditions, which also served as their own controls as soon as they were pain-free again. The ACTH and cortisol were measured before and after the TRH stimulation test. There was no significant difference in the ACTH concentration in horses with pain and the controls, between different pain intensities or between disease groups. Thus, measuring the basal ACTH concentration and performing the TRH stimulation test for the diagnosis of PPID seem to be possible in horses with a treated low to moderate pain condition.

**Abstract:**

The aim of the present study was to evaluate (i) the effects of different intensities and types of treated pain on the basal concentrations of adrenocorticotropic hormone (ACTH) and cortisol, and (ii) the thyrotropin-releasing hormone (TRH) stimulation test, to determine whether treated pain caused a marked increase of ACTH, which would lead to a false positive result in the diagnosis of pituitary pars intermedia dysfunction (PPID). Methods: Fifteen horses with treated low to moderate pain intensities were part of the study. They served as their own controls as soon as they were pain-free again. The horses were divided into three disease groups, depending on their underlying disease (disease group 1 = colic, disease group 2 = laminitis, disease group 3 = orthopedic problems). A composite pain scale was used to evaluate the intensity of the pain. This pain scale contained a general part and specific criteria for every disease. Subsequently, ACTH and cortisol were measured before and after the intravenous application of 1 mg of TRH. Results: There was no significant difference in the basal or stimulated ACTH concentration in horses with pain and controls, between different pain intensities or between disease groups. Descriptive statistics, however, revealed that pain might decrease the effect of TRH on the secretion of ACTH. There was an increase of ACTH 30 min after TRH application (*p* = 0.007) in the treated pain group, but this difference could not be statistically confirmed. Measuring the basal ACTH concentration and performing the TRH stimulation test for the diagnosis of PPID seem to be possible in horses with low to moderate pain.

## 1. Introduction

Pituitary pars intermedia dysfunction (PPID) is one of the most common endocrine diseases of older horses and ponies [1]. Hyperplasia occurs, leading to the formation of adenomas in the pars intermedia of the pituitary gland in the course of the disease. The resulting increased release of peptides of the pars intermedia, including the adrenocorticotropic hormone (ACTH), leads to the clinical characteristics of PPID [2]. Diagnosis is based on the measurement of an elevated ACTH level in the blood or the thyrotropin-releasing hormone (TRH) stimulation test, in which plasma ACTH is determined after the injection of TRH [3]. A frequent and very painful concomitant disease of PPID is laminitis [4], which, in many cases, is caused by an endocrine disease [5]. Since the pain of this disease can lead to a release of the stress hormones ACTH and cortisol, there is often uncertainty whether a horse suffering from pain can be tested for PPID, or whether the pain would lead to a false positive test result due to an increase in the ACTH value. However, since laminitis is a serious disease and often necessitates euthanasia, it is important to determine the cause early on in order to start adequate therapy. The aim of the present study was, therefore, to investigate the effects of pain on basal ACTH and cortisol levels and the TRH stimulation test in endocrinologically healthy horses.

## 2. Materials and Methods

Ethical Statement—Ethical approval was given by the LAGeSo (Landesamt für Gesundheit und Soziales), Berlin, Germany, under the number G 0056/19, according to the German animal welfare laws.

### 2.1. Horses

Seventeen horses with treated pain conditions were examined within the framework of the present study, which took place at the equine clinic of the Free University of Berlin during its normal operation. Only animals with an age of ≤15 years, which did not suffer from PPID and showed no clinical signs such as hypertrichosis, hyperhidrosis, abnormal fat redistribution, pendulum belly and/or swayback, were selected.

All pain patients examined had already spent at least one day in the clinic. After the pain had subsided, the patient was examined again and served as its own control. The horses were divided into three different groups according to their diagnosis. Group 1 included colic patients (*n* = 3), group 2 had horses with laminitis (*n* = 5) and group 3 included horses with other orthopedic conditions (*n* = 9). If some horses required surgical treatment, samples were collected for the study no less than 12 h after surgery. Due to their illnesses, all horses received different medications according to their suffering, including on the day the samples were taken. The medications were antibiotics (amoxicillin and gentamycin), anti-inflammatory drugs (flunixin), permanent drip infusions with ringer-lactate and, if necessary, also lidocaine. If the horses had to undergo surgery, sedation and anesthesia with azepromazine, xylazine and butorphanol, ketamine and diazepam, and isoflurane and dobutamine were used. The study was always performed at least three hours after administration of the medication, except for one colic patient.

### 2.2. Pain Assessment

All clinical examinations were performed by the same experienced clinician after documenting the patient’s general information (including name, breed, age, gender, weight and height, previous medication and dosage). After measuring the clinical parameters, the horses were observed for a total of five minutes and their behavior was evaluated. A pain assessment was then performed using a composite, multifactorial pain scale (Table 1, Table 2, Table 3 and Table 4) to determine the intensity of the pain The general part (Table 1) was used in all horses, while different additional parameters were assessed in patients with colic (Table 2), laminitis (Table 3) and other orthopedic problems (Table 4). The pain scales (Table 2 and Table 3) were modified according to Bussières et al. [6], Graubner et al. [7] and Rietmann et al. [8]. Obel scale (Table 3) was translated and modified according to Obel [9]. Classification of degrees of lameness was adapted by the American Association of Equine Practitioners (Table 4).

For the lameness examination, the horses were assessed in walk and trot on a straight line. They were scored with values from 0 to 3, where 0 means a physiological state and 3 means the greatest possible modification of the parameter examined in the presence of pain. Individual criteria were graded even more, i.e., up to a maximum of 5. The score obtained in this way was expressed as a percentage of the maximum score to be achieved in order to be able to compare the individual pain intensities. The different percentage limits are summarized in Table 5.

Horses were divided into four groups according to their pain intensity. Group 1 included all horses with mild (slight) pain (21–40%), group 2 had those with moderate (moderate) pain (41–60%), group 3 included all horses with severe (high) pain (61–80%) and group 4 had horses with the highest (extreme) pain.

#### 2.2.1. Adrenocorticotropic Hormone (ACTH) and Cortisol Measurements

Basal concentrations of ACTH and cortisol (0 value = ACTH 0) were measured after the pain evaluation using a precooled 4 ml potassium EDTA tube (Sarstedt, Nürnbrecht, Germany), immediately cooled and centrifuged within 10 min to determine the ACTH. The plasma obtained was pipetted into a cryogenic tube (Th. Geyer, Renningen, Germany) and frozen at −30 °C until being dispatched to the laboratory. A quantity of 36 pg/mL [2] was taken as the upper limit for ACTH 0 and 35 pg/mL as the upper limit for ACTH 30 [6]. Samples were only taken in the months of November to July to rule out the seasonal fluctuations of ACTH.

Cortisol was measured from serum samples. Consequently, blood was filled into a 10 ml serum tube (Sarstedt, Nürnbrecht, Germany) and centrifuged after coagulation. Freezing was performed in the same way as for plasma. Baseline Measurements of ACTH and cortisol were followed by the TRH stimulation test, in which both parameters (ACTH 30 or cortisol 30) were determined again 30 min after the intravenous application of a total dosage of 1 mg TRH per horse (TRH Ferring injection solution 0.2 mg/mL, Ferring Arzneimittel, Kiel, Germany).

Differences, if any, in ACTH increase (measured as the difference of ACTH 30 minus ACTH 0 concentrations) after TRH stimulation between the different pain groups were analyzed.

#### 2.2.2. Control Examination

A control examination was performed as soon as the horses were free of pain. The average time interval between pain and control blood collection was 17 ± 24 days. Two horses were discharged before they were completely pain free; therefore, the control examination in these horses had to take place in their home stables. If a horse had to be euthanized due to its underlying disease before a control examination was possible, it was not included in the calculations for the comparison between the pain condition and control, and thus, in the control group.

#### 2.2.3. Statistics

The IBM^®^ SPSS Statistics Version 23 computer statistics program (SPSS Inc., Chicago, IL, USA) was used for the statistical analysis. The statistically significant difference in the various parameters measured was calculated using the paired sample T-test. The data were not normally distributed due to the small sample size and the nonparametric Wilcoxon signed-rank test for paired samples was used. The influence of individual parameters on each other was also investigated using linear regression and the calculation of the Pearson and Spearman rank correlation coefficient, where appropriate. The Mann-Whitney U test was used to compare different parameters and the results were given as a median. If the respective parameters were to be examined in different groups, the Kruskal-Wallis test was used.

The significance level was set at 0.05, with values of *p* < 0.05 being significant, *p* < 0.01 very significant and *p* < 0.001 highly significant.

## 3. Results

### 3.1. Horses

Fifteen the 17 horses suffering from pain examined were ultimately included in the study. One colic patient was excluded from the calculations due to extreme values in the pain scale and hormonal levels. One horse suffering from laminitis developed clinical signs of PPID during the study and was, therefore, also excluded. Six of the 15 horses had to be euthanized during their stay in the clinic due to their underlying disease before they were pain-free and a control examination was possible. Therefore, only nine horses could be included in the control group (free of pain). The study population consisted of ten geldings and five mares of different breeds (1 heavy horse, 10 Warmbloods, 1 Thoroughbred and 3 ponies) with an average size of 159 ± 23 cm (109–184 cm) and an average weight of 484 ± 160 kg (184–750 kg). The average age was 10 ± 3 years (5–15 years).

Two of the horses suffered from colic (cause: incarcerated inguinal hernia, surgical therapy required or cecal gassing, conservative therapy sufficient), four were diagnosed with laminitis and nine had an orthopedic disease. Three of the four horses with laminitis were affected in both front limbs and one had laminitis in all four limbs. The orthopedic conditions included fractures (*n* = 3), arthrosis (*n* = 2), septic arthritis, osteochondrosis dissecans, phlegmons and a desmopathy of both suspensory ligaments (*n* = 1 each). Five horses from this group were operated on due to their disease.

The control group included five geldings and four mares of different breeds with an average height of 151 ± 25 cm (109–175 cm), an average body weight of 426 ± 163 kg (184–650 kg) and an average age of 10 ± 3 years (5–14 years).

### 3.2. General Clinical Examination

Only the rectally measured internal body temperature was significantly different between sick and control horses (*p* = 0.012) in the general clinical examination. In the sick state, the median temperature of 37.9 °C was generally higher than the median temperature of healthy horses at 37.6 °C.

### 3.3. Adrenocorticotropic Hormone (ACTH) Concentration

None of the horses included in the study exceeded the reference range of 36 pg/mL established by the laboratory. The 0 value for ACTH in the control group (*n* = 9) was, on average, higher in the horses with pain (mean 13.6 ± 6.8 pg/mL) than during the control examination (mean 10.7 ± 6.5 pg/mL). However, this difference was not significant (Table 6).

#### 3.3.1. Adrenocorticotropic Hormone 30 Minutes (ACTH 30) after Thyrotropin-Releasing Hormone (TRH) Stimulation Test

ACTH was measured before (ACTH 0) and 30 min after TRH application (ACTH 30). There was a significant increase (*p* = 0.007) in ACTH 30 (mean value 19 ± 10.6 pg/mL) after the application of TRH compared to the 0 value (mean value 12.8 ± 6.2 pg/mL) in the horses exposed to pain (*n* = 15). The ACTH 30 value in the horses suffering from pain was, thus, above the upper limit value of 35 pg/mL. The average increase in ACTH after TRH stimulation during pain was lower than at the control examination (3.4 pg/mL median vs. 8.9 pg/mL median). However, this difference was not significant (Table 6). 

There was also a significant increase (*p* = 0.010) in ACTH 30 (mean 22.4 ± 14.5 pg/mL) compared to the 0 value (mean 10.7 ± 6.5 pg/mL) in the control horses (*n* = 9). Each horse showed an increase in ACTH 30 min after TRH application. The ACTH 30 value rose to 52.9 pg/mL in one of the control horses, which was above the upper limit of 35 pg/mL. The average ACTH 30 value during the painful condition in the control group, however, was slightly lower overall (mean value 19.5 ± 12.8 pg/mL) than at the control examination (mean value 22.4 ± 14.5 pg/mL). This difference was not significant (Table 6). 

#### 3.3.2. Pain Groups and Adrenocorticotropic Hormone (ACTH) Levels

Ten of 15 horses in the study population suffered from severe pain (group 1), and five had moderate pain (group 2). The ACTH 0 was higher, on average, in group 1 (13.7 ± 6.8 pg/mL) than in group 2 (10.9 ± 4.9 pg/mL). However, this was not significant. The ACTH 30 was also higher, on average, in group 1 (21 ± 11 pg/mL) than in group 2 (15 ± 9.8 pg/mL), but this difference could not be statistically confirmed.

Furthermore, there were no significant differences in the 0 and 30 values of the ACTH concentration measured between the disease groups formed (colic, laminitis, orthopedic diseases).

### 3.4. Cortisol Concentration

The median of the cortisol basal value (cortisol 0 value) of all horses exposed to pain (*n* = 15) was 30 ng/mL and, thus, at the lower limit of the reference range for cortisol of 30–130 ng/mL given by the laboratory. An increase of cortisol 0 could be determined overall, depending on the pain score. However, this was not statistically significant. The median for the horses with pain in the control group (*n* = 9) was 31 ng/mL. The median was 24 ng/mL during the control examination. 

There was a significant increase (*p* = 0.022, Wilcoxon test) in cortisol 30 compared to the 0 value 30 min after the application of TRH in the horses examined that were exposed to pain (*n* = 15). The median of cortisol 30 was 51.3 ng/mL (Table 6).

There was also a significant increase (*p* = 0.004, Wilcoxon Test) in the cortisol concentration 30 min after TRH stimulation in the nine horses of the control group. The mean value was 37.8 ± 13.7 ng/mL and the mean value of the 0 value was 24.3 ± 8.6 ng/mL (Table 6).

A higher cortisol 30 value could be measured during the pain state than during the control examination (median 50.8 ng/mL vs. 31.3 ng/mL) in the control group. However, this difference was not statistically significant.

### 3.5. Pain Groups and Cortisol Values

The 0 value in group 1 was higher (median of 30.5 ng/mL, range 21–89.5 ng/mL) than in group 2 (median of 27.0 ng/mL, range 15.2–103 ng/mL); however, this difference was not statistically significant. By contrast, cortisol 30 was higher in group 2 (median of 53.9 ng/mL, range 30.2–116 ng/mL) than in group 1 (median of 48.2 ng/mL, range 31.9–116 ng/mL; statistically unsubstantiated). Looking more closely at the different disease groups, the highest values were measured in horses with colic, followed by those with laminitis. The horses with an orthopedic disease had the lowest values. This result is not statistically significant. The same applies to the value of cortisol 30.

## 4. Discussion

It has long been known that pain leads to an activation of the hypothalamic-pituitary-adrenal (HPA) axis, resulting in the release of ACTH and cortisol from the pars distalis of the pituitary gland [10,11]. The measurement of baseline ACTH for PPID diagnosis in horses in severe pain is, therefore, not recommended due to the possibility of false positive results [5,12,13,14]. However, it is unclear at what intensity the pain can provoke such a release and whether the TRH stimulation test is influenced as well. The application of TRH directly stimulates melanotropes in the pars intermedia of the pituitary gland to release ACTH. It is not known to what degree these cells might be influenced by stress and pain, as they are not usually involved in the stress response. Glucocorticoid receptors, mediating the stress response, have been found in the pars intermedia of the pituitary gland in rats, suggesting a potential responsiveness of melanotropes to stress [15,16,17,18,19]. The present study was intended to measure the ACTH and cortisol levels in endocrinologically healthy horses in different pain states. Therefore, it had to be ensured that a PPID in which the ACTH concentration was elevated due to disease could be excluded beforehand. Thus, only those horses which were younger than 15 years of age and did not show clinical symptoms of PPID were included in the study. This left only a small residual risk of hyperplasia of the pars intermedia in the patient population, as the disease occurs predominantly in older animals [2].

Treated pain, ranging from extreme to moderate, did not result in a significant increase in blood ACTH levels in the present study. The ACTH basal value did not generally exceed 36 pg/mL in any of the horses exposed to pain, and the stimulated ACTH value did not exceed 35 pg/mL in any horse.

Previous studies have preferred to measure cortisol to assess the effects of pain on stress hormones [12,20,21,22,23]. Ayala et al. [24] investigated the effects of various diseases on ACTH levels in horses. They evaluated horses with laminitis, colic, acute conditions (e.g., trauma or arthritis), chronic conditions (e.g., osteoarthritis) and horses neutered under general anesthesia. The ACTH levels were significantly higher than in healthy control animals in all disease groups except neutered horses. However, the severity of the respective disease or the pain associated with it was not separately assessed in the study. Nevertheless, it can be assumed that the diseases were associated with some degree of pain. The highest values were found in horses with laminitis, colic and acute diseases. They were so highly elevated that a false positive result would have been obtained in PPID diagnostics. The reason for this could be that in the study by Ayala et al. [24], the horses were not treated with analgesics before the measurements were taken. In addition, the samples were taken only 30–60 min after arrival at the clinic. Transport, a new environment and veterinary examinations are also stressors for the horse; therefore, these may have contributed to the increase in ACTH secretion [25,26,27]. However, the results of the present study did not show any difference in ACTH concentrations in horses with different diseases, nor did the concentrations of ACTH differ significantly between pain groups with different pain intensities. This is in contrast to the results of Niinistö et al. [28] and Towns et al. [29], who found that the severity of colic and the associated pain intensity correlated significantly with the ACTH values, whereas average ACTH concentrations of horses with slight and moderate colic were within the reference range and did not differ significantly from healthy control animals, as was the case in the present study. A direct comparison of the two studies is hardly possible, since the pain in the present study was not only of abdominal origin, but horses mainly with pain caused by laminitis or an orthopedic disease were also examined. It must also be critically questioned whether systemic effects, such as hypovolemic shock, cytokines or endotoxemia, may have influenced the activity of the HPA axis during the occurrence of colic [24,30].

Three animals with laminitis in the present study were found to have had a lower ACTH value during the painful period than in the control study. It is possible that continued stimulation of the HPA axis may have caused system exhaustion, resulting in a decrease in stress hormones [26,27,28]. The results of Ayala et al. [29], who found significantly higher ACTH levels in horses with chronic diseases than in healthy control animals, speak against this. The pain in the present results was considered as a snapshot. It is, therefore, unclear whether some horses had been suffering from pain for a long time and whether the chronic pain led to the reduced ACTH concentrations.

As an intermediate result, it can be stated from the observations made that slight to moderate pain does not seem to significantly influence ACTH secretion in horses. Strong exposure to stressors (such as pain) is required to cause a significant increase in the ACTH concentration [29,31,32,33,34]. The measurement of ACTH levels for PPID diagnosis would, thus, be possible for slight to moderate pain.

All horses suffering from pain were given analgesics on the day of sampling. Nonsteroidal anti-inflammatory drugs, such as flunixin meglumine, phenylbutazone and metamizole, have different half-lives, which may still have been effective at the time of sampling. The application of flunixin meglumine in previous studies showed no influence on the ACTH or cortisol concentration [20,23,35,36]. However, it is possible that ACTH secretion is inhibited by the administration of opioids [37,38].

General anesthesia, as performed in some of the patients in this study, could also affect the ACTH and cortisol. General anesthesia leads to a stress response of the body, which results in an increase in the ACTH and cortisol [39]. However, a continuous drip infusion with α2 agonists may reduce the stress response during anesthesia [40]. A balanced anesthesia with isoflurane and an infusion with xylazine was also performed in the present study. In addition, the horses received a xylazine bolus shortly before being brought to the recovery box. It is, therefore, possible that the stress reaction to the anesthesia was alleviated in this way. In a study by Taylor [39], the ACTH and cortisol levels fell back to their initial values just 4 to 6 h after halothane anesthesia without surgical intervention. The ACTH and cortisol values in the present study were determined at least 12 h after getting up from anesthesia to achieve a sufficient time interval regarding the possible effects of anesthesia. The elimination half-life of isoflurane is low because it is barely metabolized; rather, it is almost completely exhaled via the lungs and, additionally, has a lower solubility in blood and fat [41]. Xylazine administered at the end of anesthesia should also no longer show effective plasma levels after 12 h. It is, therefore, unlikely that the drugs used for anesthesia in the present study had an effect on the stress hormones.

The administration of lidocaine in a continuous drip after surgical intervention cannot be conclusively assessed regarding the release of stress hormones, as it has a predominantly anti-inflammatory effect and, thus, contributes to the reduction of pain [42,43]. By contrast, the effects of α2 agonists on the concentrations of ACTH and cortisol are discussed contradictorily in the literature. Pakkanen et al. [44] found increased ACTH and cortisol levels 15 min after the application of Romifidine. However, these levels dropped back to their initial values within 60 min. Other studies reported constant or decreasing cortisol concentrations [45,46]. There are currently no studies on the effects of heparin, omeprazole and antibiotics on stress hormones in horses.

The pain scale used in this study to evaluate the pain intensity of horses was a composite, multifactorial scale. The use of such systems has proven to be effective for pain assessment [6,47]. While one-dimensional scales are either based on a purely subjective assessment of pain intensity or depend very much on the clinical experience of the user, a multifactorial scale provides a more objective evaluation of pain by assessing clinical criteria together with clearly defined behavioral changes [48,49]. The animals in the present study suffered from different types of pain. Horses show different behavior in different pain conditions [50]. A reliable pain scale should, therefore, contain parameters that are specific to the type of pain in question [49,51]. Consequently, a pain scale was used in the present study which included a general part for all animals exposed to pain and, additionally, specific parameters for horses with visceral pain (colic), laminitis-associated pain and orthopedic pain. A limiting factor might be the short observation period for pain evaluations. A longer observation period would not have been practical as the pain intensities in this study were evaluated under clinical conditions. 

Only the internal body temperature differed clinically significantly between the pain state and control in the present study. May [51] even found a direct correlation between the body temperature and a subjectively assessed degree of pain. Bussières et al. [6], on the other hand, described internal body temperature as a nonspecific criterion for pain. In the present study, an increased motor activity due to the active pain behavior or the stress associated with the pain may have led to the increase in body temperature observed. It should be noted, however, that the internal body temperature was also within a physiological range in the pain patients.

While performing the TRH stimulation test, ACTH can be sampled 10 or 30 min after TRH administration, although 10 min has recently been recommended. In the present study, blood for post-TRH ACTH was collected 30 min after TRH injection due to handling reasons, since there was only one investigator, and blood samples had to be processed immediately after collection. The intravenous application of TRH resulted in a significant increase in ACTH concentrations in the horses in the present study during the painful period and at the follow-up examination. This confirms the results of McFarlane et al. [52] and Beech et al. [10]. However, it was striking that the ACTH level decreased after TRH stimulation in 3 of the 15 horses suffering from pain. A control examination could also be performed on these three horses, allowing a direct comparison to be made of the ACTH 30 values in pain and the control examination. During the latter, these three horses again showed an increase in ACTH concentrations after TRH stimulation. This was the case for all horses during the control examination. Despite a lack of significance, descriptive statistics also showed that the average ACTH 30 value during the pain state tended to be lower than during the control examination. The increase in ACTH after TRH stimulation was also lower in horses in pain than in the control examination, but not significantly. The reason for the partly missing stimulating effect of TRH and the low ACTH 30 values during the pain state could be a missing response of the pituitary gland to TRH. Similar effects have been described in nonthyroidal illness syndrome in adult horses and in foals, where different conditions, such as stress, pain and sepsis, alter all levels of the HPT axis. In more detail, cytokines, such as IL1-β, IL-6 and TNF-α, induce a hypothalamic and pituitary dysfunction and suppress TSH, T_4_ and T_3_ synthesis and secretion in proportion to the disease severity [53,54,55,56,57,58,59]. The effects of pain on the pituitary reaction to TRH stimulation have not yet been investigated in horses, but alterations comparable to those seen in nonthyroidal illness syndrome seem likely, at least in critically ill patients. In humans, some patients with chronic back pain showed a reduced response of the pituitary gland to TRH [51]. The authors suspected a possible downregulation of TRH receptors due to the chronic hypersecretion of TRH. However, it is unclear whether pain in horses could lead to a hypersecretion of TRH and, consequently, to a downregulation of the TRH receptors in the pars intermedia. It also remains questionable to what extent the drugs administered could have influenced the TRH stimulation test. 

Low to moderate pain did not lead to a significant increase in cortisol. Neither an influence of pain intensity on cortisol nor a difference between horses with slight and moderate pain could be found. Raekallio et al. [54] and Rietmann et al. [8] could not prove a correlation in horses after orthopedic surgery or in horses with laminitis. In another study, cortisol did not correlate with postoperative pain after arthroscopy either [20]. By contrast, Pritchett et al. [18] described significantly increased cortisol concentrations in horses with postoperative pain after laparotomy. Horses with acute laminitis also showed significantly increased cortisol levels [51]. Bussières et al. [6] and May [51] even showed a positive correlation between cortisol and pain intensity. It is likely that the strength of cortisol secretion depends on the intensity of pain. The horses in the present study suffered from slight to moderate pain. However, it was described that cortisol increases significantly only in severe pain [54]. Raekallio et al. [21] also noted in their study that the pain after arthroscopy may not have been strong enough to increase cortisol concentrations significantly. By contrast, Pritchett et al. [23] examined horses after laparotomy, and Bussières et al. [6] examined horses with acute synovitis, both of which may have been associated with a higher degree of pain. Niinistö et al. [28] could only measure significantly increased values in horses with severe colic symptoms compared to healthy control animals. However, a direct comparison is difficult because the pain intensities in the respective studies were always evaluated in different ways.

Horses with colic showed the highest cortisol levels in the present study, followed by those with laminitis and orthopedic patients. The results suggest that the origin of the pain may influence the secretion of cortisol. Ayala et al. [24] showed higher cortisol concentrations in horses with colic, laminitis or acute conditions, such as trauma, than in those with chronic conditions, such as osteoarthritis. The patients with colic in the present study maybe showed stronger pain than those from other groups, or endotoxemia and/or cardiovascular disorders, which are additional stressors and activate the HPA axis [55].

Regarding the cortisol concentration, two horses in the study population showed a lower cortisol value during the painful condition than during the control examination. The duration of pain may have played a role in this, as has already been considered when discussing the ACTH value. Whether there is an increase or decrease in cortisol in chronic pain is discussed differently in the literature [60,61,62,63]. Investigations in horses with prolonged pain have mostly revealed lowered cortisol concentrations. Cortisol initially increased significantly in horses with experimentally induced lameness, but then decreased again within a few hours, although the lameness persisted [35]. This could have been due to the circadian rhythm of cortisol, with lower values in the evening [64]. Moreover, Mills et al. [65] reported that the induction of chronic inflammation led to a gradual decrease in cortisol concentration.

A cortisol response varies greatly from individual to individual and can be influenced by many factors [57,64,65,66]. The measurements in the present study were always performed twice on the same animal, at the same time of day, by the same investigator and, if possible, in the same environment in order to minimize many factors influencing cortisol. However, it is likely that the type, intensity and duration of pain, which was different in each horse, and the different diseases associated with the pain, influenced the results in the present study. This makes it almost impossible to determine the influence of pain on cortisol secretion. 

Measuring cortisol in response to TRH is not a diagnostic test for PPID. It was only quantified to evaluate the influence of pain and stress. A significant increase in cortisol concentrations occurred 30 min after the intravenous application of TRH in the present study. This was the case during both the pain state and the control examination. However, the cortisol level in four of the horses suffering from pain decreased after TRH stimulation. Three of these four horses also showed decreased ACTH values after the application of TRH. The latter stimulates the release of ACTH from the pituitary gland; ACTH, in turn, causes the secretion of cortisol from the adrenal cortex, which explains the parallel decrease in values [48]. There was also a decrease in cortisol after TRH stimulation in one horse during the control examination. These results reflect the different reactions of cortisol to TRH observed in other studies. There was either a significant increase in horses without PPID [50] or the concentrations remained approximately the same [67,68]. No significant difference in cortisol 30 values between the pain state and the control examination could be found. 

## 5. Conclusions

The results of the present study show that slight to moderate pain with a duration of >24 h does not significantly affect the basal ACTH and cortisol levels or TRH stimulation test in endocrinologically healthy horses treated with pain medication. The measurement of ACTH and the performance of the TRH stimulation test for PPID diagnostics can, therefore, be performed on horses in pain as long as they are not suffering from massive pain or showing a significantly disturbed general condition. It might, however, be possible that pain induces a diminished response to TRH stimulation in individual animals.

Since a uniform gold standard for assessing pain intensity has not yet been developed, it would first be necessary to develop a generally applicable pain assessment scale for different types of pain in horses to be able to differentiate clearly between different pain intensities.

Further studies on a greater number of horses with uniform pain patterns are also necessary to determine the factors influencing the stress hormones even more precisely. In addition, more studies on the effects of pain and the influence of medication on the TRH stimulation test are still lacking.

## Figures and Tables

**Table 1 animals-10-01426-t001:** Pain scale to determine the intensity of the pain. General Part.

Parameter	Importance/Symptoms	Score
Behavior Movements (spontaneous)	Horse stands relaxed or shows calm movements.	0
	Reduced movement or mild agitation.	1
	Reluctance to move or moderate agitation.	2
	Does not move, appears introverted or makes uncontrollable forward movements.	3
Appetite Feed intake	Eats hay willingly (may wear a muzzle).	0
	Eats hay reluctantly.	1
	Shows little interest in hay, eats only a few stalks or takes hay in the mouth but does not chew or swallow.	2
	Shows no interest in hay and does not eat any.	3
Sweating	No sweating, dry coat.	0
	Coat feels clammy.	1
	Coat feels damp, beads of sweat visible.	2
	Heavy sweating, sweat runs off the body.	3
Heart rate	22–44	0
	45–52	1
	53–60	2
	>60	3
Respiratory rate	<20	0
	20–24	1
	25–30	2
	>30	3
Internal body temperature	36.9–38.5 °C	0
	36.4–36.9 °C or 38.5–39.0 °C	1
	35.9–36.4 °C or 39.0–39.5 °C	2
	35.4–35.9 °C or 39.5–40.0 °C	3

**Table 2 animals-10-01426-t002:** Pain scale to determine the intensity of the pain. Additional parameters for horses with colic.

Parameter	Importance/Symptoms	Score
Gut sounds	Normal motility (++)	0
	Reduced motility (+)	1
	No motility (-)	2
	Hypermotility (+++)	3
Kicking to the stomach	Horse stands still, no kicking to the stomach.	0
	Occasionally kicks against the stomach. (1–2 times in 5 min)	1
	Kicks regularly against the stomach. (3–4 times in 5 min)	2
	Kicks excessively against the stomach. (>5 times in 5 min)	3
Pawing	Horse stands still, no scratching.	0
	Occasional scratching. (1–2 times in 5 min)	1
	Regular scratching. (3–4 times in 5 min)	2
	Excessive scratching. (>5 times in 5 min)	3
Head movements	No sign of discomfort, head is mainly held straight in front of the body.	0
	Intermittent, lateral or vertical head movements, occasionally looking at the flank (1–2 times in 5 min) and/or lifting the lips (1–2 times in 5 min).	1
	Intermittent, violent, lateral or vertical head movements, looking regularly at the flank (3–4 times in 5 min) and/or lifting the lips (3–4 times in 5 min).	2
	Continuous head movements, looking excessively at the flank (>5 times in 5 min) and/or lifting the lips (>5 times in 5 min).	3
Lying down, rolling	Horse stands quietly in the box.	0
	Occasionally lying down.	1
	Regularly lying down and getting up again, rolling.	2
	Horse repeatedly throws itself down uncontrollably and rolls on the ground.	3

**Table 3 animals-10-01426-t003:** Pain scale to determine the intensity of the pain. Additional parameters for horses with laminitis.

Parameter	Importance/Symptoms	Score
Posture	Normal movements, stands still with even weight distribution on all four limbs.	0
	Occasional weight shift with temporary relieving posture, slight muscle tremor.	1
	Abnormal weight distribution, relieves a limb.	2
	Muscle tremor, exhaustion, sawhorse posture/arched back.	3
Obel scale	No abnormalities in movement.	0
	At rest, constant shifting of weight from one leg to the other. No lameness at walking pace but the horse shows a shortened, stiff walk at a trot. (Obel grade I)	1
	Horse walks willingly at walking pace but with a noticeably shortened and stiff gait. A limb can be lifted without any problems. (Obel grade II)	2
	Horse moves only reluctantly. A limb is difficult or impossible to pick up. (Obel Grade III)	3
	Horse refuses to move. Only moves when forced to. (Obel Grade IV)	4
Pulsation of the Aa. digitalis palmaris lateralis and medialis	Physiological pulsation.	0
	Slightly increased pulsation.	1
	Moderately increased pulsation.	2
	Highly increased pulsation.	3
Reaction to the hoof pincers	No retraction of the limb.	0
	Retraction of the limb when strong pressure is applied with the hoof pincers.	1
	Retraction of the limb, even under slight pressure.	2
	Retraction of the limb, even when pressure is exerted only with the hand.	3

**Table 4 animals-10-01426-t004:** Pain scale to determine the intensity of the pain. Additional parameters for horses with an orthopedic disease.

Parameter	Importance/Symptoms	Score
Posture	Normal movements, stands still with even weight distribution on all four limbs.	0
	Occasional weight shift with temporary relieving posture, slight muscle tremor.	1
	Abnormal weight distribution, relieves a limb.	2
	Muscle tremor, exhaustion, sawhorse posture/arched back.	3
Degree of lameness	No lameness discernible. (Grade 0/5)	0
	Lameness is difficult to detect and not continuously visible. (Grade 1/5)	1
	The lameness is difficult to detect on a straight line at walking pace and trot. Under certain conditions, however, it can be continuously detected (e.g., on the circle, on hard ground). (Grade 2/5)	2
	Lameness continuously visible at the trot, under all conditions. (Grade 3/5)	3
	Severe lameness, obvious at a walking pace. (Grade 4/5)	4
	Severe lameness, only short-term loading of the limb in motion and/or at rest or complete relief of the limb. (Grade 5/5)	5
Reaction when lifting the contralateral limb	Contralateral limb can be lifted without problems.	0
	Contralateral limb can only be lifted with difficulty and for a short time.	1
	Contralateral limb cannot be lifted at all.	2

**Table 5 animals-10-01426-t005:** Significance of the individual percentage values compared to the individual pain intensities.

Percentile Ranks (%)	Significance
0–20	Nonexistent
21–40	Slight
41–60	Moderate
61–80	High
81–100	Extreme

**Table 6 animals-10-01426-t006:** TRH stimulation test of horses treated for painful conditions.

Horse (Number)	Disease State	Pain Score (%)	Medication (Hours between Administration and Blood Sampling)	ACTH 0 (pg/mL)	ACTH 30 (pg/mL)	Cortisol 0 (ng/mL)	Cortisol 30 (ng/mL)
1	Orthopedic	50	Phenylbutazone (3.5) Amoxicillin (3)	6.5	8.64	27	-
2 *	Orthopedic	25	Phenylbutazone (7.5)	6.8	10.2	29	51.7
3	Colic	55.5	Amoxicillin (6.25) Heparin (6.25) Metamizole (4.25) Flunixin (3.25) Lidocaine (3.25)	18.6	22.7	103	116
4	Orthopedic	35.71	Amoxicillin (7) Gentamicin (7) Flunixin (7) L-Polamivet (4)	9.1	15.1	21	31.9
5	Orthopedic	42.85	Phenylbutazone (6.5)	9.4	15.2	23	53.3
6 *	Orthopedic	39.28	Amoxicillin (6) Gentamycin (6) Flunixin (6)	12	25.8	67	116
7 *	Laminitis	25.8	Phenylbutazone (6.5)	14.3	33.4	29	41.3
8 *	Laminitis	25.8	-	24.8	40.2	30	43.5
9	Orthopedic	25	Phenybutazone (5.75)	18.5	29.4	25	45.5
10 *	Laminitis	48.38	Flunixin (5.75) Heparin (5.75)	7.4	2.5	60	54.4
11 *	Orthopedic	28.57	Amoxicillin (5.25) Gentamycin (5.25) Phenylbutazone (5.25) Omeprazole (5.25)	17.9	13.1	31	35.2
12 *	Laminitis	35.48	Flunixin (5.25) Heparin (5.25)	5	5.5	58	50.8
13 *	Orthopedic	32.14	Amoxicillin (5.75) Dembrexin (5.75) Flunixin (3.75)	21.62	18.71	68.5	62.4
14	Colic	33.33	Amoxicillin (7.25) Gentymycin (7.25) Heparin (7.25) Metamizole (1.25) Xylazin (1.25) Butorphanol (0.75) Flunixin (0.75)	6.81	18.3	89.5	77
15 *	Orthopedic	46.42	-	12.81	26.2	15.2	30.2

ACTH = Adrenocorticotropic hormone, TRH = Thyrotropin-releasing hormone; * A control examination was possible.

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
