# Peer review of "Can Endocrine Dysfunction Be Reliably Tested in Aged Horses That Are Experiencing Pain?"

_animals, 2020, doi:10.3390/ani10081426_

Round 1
Reviewer 1 Report
Dear authors,
Interesting study with a topic many practitioners would like to know about. Unfortunately not many horses were included in the complete study which makes it difficult to interpret the data since the 9 horses were divided over 3 different disease groups and 2 different pain score groups. Findings are therefore often indicative but not significantly different. The findings on the TRH response are new, very promising but not significant. This deserves more attention in future research.
Is it somehow possible to add some extra cases? It would really make the study much stronger although I realize this is something difficult to ask.
Line 121 - what was the upper limit for the ACTH0? Was there a difference in season when samples were taken? Please mention the months in which data was collected.
Could you explain why you took the 30 minute ACTH and cortisol and not the generally preferred 10 min sampling? Maybe it is wise to add a note in the discussion that TRH stimulation test with measuring cortisol is not relevant for diagnosing PPID and only used here to look at the influence of stress. The ACTH and cortisol levels are not longer correlated due to the probably not biological active ACTH which is secreted by the pars intermedia in PPID horses.
Line 152-153 - were the extreme values in the pain scale score or the hormonal levels? Please explain a bit more why you took those horses out.
Line 165-167 - consider to use one term for the measurements during the painfree period. Now it is mentioned control group and comparison group, this is a bit confusing for the reader.
Figure 2 - delete p*=…..
Line 223-229 - could you give the range of the median levels.
Line 342-347: I think that there is a lot of information regarding the effect of illness on the thyroid responses, think of NTIS (non-thyroidal illness syndrome). Acute diseases like sepsis, trauma, surgery all suppress thyroid hormones, chronic disease seem to do the opposite in man as well as several animal models. NTIS has been described for horses as well so please add this to your discussion and relate to the responses found in your TRH test.
Line 385 - same investigator, same environment - in the material and methods there is written that control samples have been taken in some case when horses were back home. This seems to be in contradiction with the statement in line 385.
Conclusion - the conclusion should be more precise in my opinion - you are looking in your study at horse not suspected of PPID with at least more than 24 hours pain, at least one day at your clinic and treated with painkillers; under these restrictions you did not find an increase on your basal ACTH or TRH stimulation test levels.
Nevertheless I think it is still debatable whether it is advisable to test for PPID during illness by using the TRH stimulation test according to your finding of diminished response to TRH. Could this lead to false negative results in early PPID cases when horses suffer acute diseases?
In general: I think it deserves a bit more explanation in the discussion or introduction that the TRH stimulation test mostly stimulates melanocytes in the pars intermedia where GC receptors are missing and stress therefore does not influence the pars intermedia by HPA axis directly if my knowledge is correct. Basal ACTH/cortisol levels (coming from the pars distalis for 98%) are therefore expected to be more influenced by stress than the TRH stimulated response in ACTH from the pars intermedia?
Could you add an addendum with individual data of the horses? Table with Disease state, CPS, medication in relation to sampling and outcome of TRH stim. You have some horses that respond differently on an individual basis, it helps the reader to interpret this dataset by giving specific information on the different cases since the numbers are low.
Author Response
Dear reviewer,
thanks a lot for reviewing our manuscript.
It is true, that the number of horses is not so big. It is not possible to add some extra cases as we have only the permission from the LAGESO (our animal welfare committee) for the number of horses we have examined.
Line 121 - what was the upper limit for the ACTH0? Was there a difference in season when samples were taken? Please mention the months in which data was collected.
I added the upper limit for ACTH 0.
Could you explain why you took the 30 minute ACTH and cortisol and not the generally preferred 10 min sampling? Maybe it is wise to add a note in the discussion that TRH stimulation test with measuring cortisol is not relevant for diagnosing PPID and only used here to look at the influence of stress. The ACTH and cortisol levels are not longer correlated due to the probably not biological active ACTH which is secreted by the pars intermedia in PPID horses.
I have included an explanation on the use of the 30 minutes sampling, and also a note in the discussion, that the TRH stimulation test with measuring cortisol is not a diagnostic test for PPID.
Line 152-153 - were the extreme values in the pain scale score or the hormonal levels? Please explain a bit more why you took those horses out.
I added a more detailed explanation.
Line 165-167 - consider to use one term for the measurements during the painfree period. Now it is mentioned control group and comparison group, this is a bit confusing for the reader.
I changed this.
Figure 2 - delete p*=…..
I added the p-value.
Line 223-229 - could you give the range of the median levels.
I included the range of the median levels.
Line 342-347: I think that there is a lot of information regarding the effect of illness on the thyroid responses, think of NTIS (non-thyroidal illness syndrome). Acute diseases like sepsis, trauma, surgery all suppress thyroid hormones, chronic disease seem to do the opposite in man as well as several animal models. NTIS has been described for horses as well so please add this to your discussion and relate to the responses found in your TRH test.
Thank you for this important comment. I have added a short part about NTIS.
Line 385 - same investigator, same environment - in the material and methods there is written that control samples have been taken in some case when horses were back home. This seems to be in contradiction with the statement in line 385.
I have adjusted the part about control examinations in the material and methods.
Conclusion - the conclusion should be more precise in my opinion - you are looking in your study at horse not suspected of PPID with at least more than 24 hours pain, at least one day at your clinic and treated with painkillers; under these restrictions you did not find an increase on your basal ACTH or TRH stimulation test levels. Nevertheless I think it is still debatable whether it is advisable to test for PPID during illness by using the TRH stimulation test according to your finding of diminished response to TRH. Could this lead to false negative results in early PPID cases when horses suffer acute diseases?
I have included more information in the conclusion and added a part about a possibly diminished TRH response.
In general: I think it deserves a bit more explanation in the discussion or introduction that the TRH stimulation test mostly stimulates melanocytes in the pars intermedia where GC receptors are missing and stress therefore does not influence the pars intermedia by HPA axis directly if my knowledge is correct. Basal ACTH/cortisol levels (coming from the pars distalis for 98%) are therefore expected to be more influenced by stress than the TRH stimulated response in ACTH from the pars intermedia?
Thank you for this interesting comment. We did not find studies about GC receptors in the pars intermedia in horses, but different studies in rats found GC receptors in the pars intermedia and their presence seems to be increased by the absence of dopaminergic innervation (References: Antakly and Eisen (1984) Immunocytochemical localization of glucocorticoid receptors in target cells. Endocrinology 115: 1984-1989. Antakly et al. (1985) Induced expression of the glucocorticoid receptor in the rat intermediate pituitary lobe. Science 229:277-279. Antakly et al. (1987) Tissue-specific dopaminergic regulation of the glucocorticoid receptor in the rat pituitary. Endocrinology 120: 1558-1562. Bertini et al. (1989) Glucocorticoid receptor immunoreactivity in the rat intermediate lobe. J Neuroendocrinol 1: 465-471. Seger et al. (1988) Stimulation of pro-opiomelanocortin gene expression by glucocorticoids in the denervated rat intermediate pituitary gland. Neuroendocrinol 47: 350-357.). So it seems speculative what happens to GC receptors in horses with PPID. But nevertheless we have added a few sentences on this subject.
Could you add an addendum with individual data of the horses? Table with Disease state, CPS, medication in relation to sampling and outcome of TRH stim. You have some horses that respond differently on an individual basis, it helps the reader to interpret this dataset by giving specific information on the different cases since the numbers are low.
We added the table.
Reviewer 2 Report
This is an interesting manuscript concerning a topic that is of high interest to horse owners and equine practitioners. There is one over-riding methodologic problem with the study, however. The authors did not study horses in various amounts of pain, they were studying horses that were being treated for a variety of painful conditions. What is more, they were treated with a variety of analgesics of several drug classes. This is a major confounder of the study and makes comparisons to earlier publications problematic. The fact the horse’s ACTH was the same before and after the painful condition resolved could be interpreted to mean that the analgesic that the horse had been placed on was successful in removing the pain as intended. Despite this major drawback in study design, there is value to the data presented. The situation described in this manuscript is the reality for equine practitioners as they would not allow a painful horse to suffer merely to rule out PPID. If the manuscript it re-written to reflect the fact that they are studying horses that are being treated for painful conditions rather than horses suffering from painful conditions there is merit to the study.
Abstract: It would be helpful to readers if some numerical values are given in the abstract
Line 123: Recent recommendations are to collect blood for post-TRH ACTH determination 10 minutes rather than 30 minutes after TRH administration. Authors should discuss why 30 minutes was selected.
Line 130: Horses do not typically resolve laminitis to return to normal hoof anatomy, so it is not clear what is meant by recovery from laminitis. Also, the average time interval between painful and non-painful blood collection should be measured.
Line 152: What is meant by extreme values? Having a value that is an outlier is not a sufficient reason to remove it from the data analysis.
160: What does “tethering” mean? This word has no meaning in this context in English
Table 1a
Is the movement being evaluated spontaneous movement or movement when horse is stimulated?
322: The authors should make the distinction between statistically and clinically significant
Table 1b
Gut sounds do not equate to GI motility. The first box should be renamed to either “gut sounds” or “borborygmi”
Scratching has no meaning in English in this context. Do the authors mean pawing?
Evaluating the surgical wound should not be included in this table. Post-operatively a horse should not exhibit colic and pre-surgically a horse will not have an incision.
Author Response
Dear reviewer, thanks a lot for revising our manuscript.
If the manuscript it re-written to reflect the fact that they are studying horses that are being treated for painful conditions rather than horses suffering from painful conditions there is merit to the study.
We changed this.
Abstract: It would be helpful to readers if some numerical values are given in the abstract.
I inserted the interesting p-value.
Line 123: Recent recommendations are to collect blood for post-TRH ACTH determination 10 minutes rather than 30 minutes after TRH administration. Authors should discuss why 30 minutes was selected.
I have included an explanation on the use of the 30 minutes sampling, and also a note in the discussion, that the TRH stimulation test with measuring cortisol is not a diagnostic test for PPID.
Line 130: Horses do not typically resolve laminitis to return to normal hoof anatomy, so it is not clear what is meant by recovery from laminitis. Also, the average time interval between painful and non-painful blood collection should be measured.
The sentence has been altered and average time between first and second exam added.
Line 152: What is meant by extreme values? Having a value that is an outlier is not a sufficient reason to remove it from the data analysis.
I added a more detailed explanation.
160: What does “tethering” mean? This word has no meaning in this context in English I explained this.
Table 1a
Is the movement being evaluated spontaneous movement or movement when horse is stimulated? This movement was spontaneous.
322: The authors should make the distinction between statistically and clinically significant.
We changed this.
Table 1b
Gut sounds do not equate to GI motility. The first box should be renamed to either “gut sounds” or “borborygmi”
I changed this.
Scratching has no meaning in English in this context. Do the authors mean pawing?
Yes, I changed this.
Evaluating the surgical wound should not be included in this table. Post-operatively a horse should not exhibit colic and pre-surgically a horse will not have an incision.
We deleted this.
Round 2
Reviewer 1 Report
Dear authors,
Thank you for your responses to the comments. I really appreciate Table 3, it nicely shows that the horses were on different painkillers. Thank you for that.
I just have some typographic remarks.
Table 3: Kortisol should be Cortisol
Line 139: conrol should be control; please check document on this word. It is written more often as conrol.
Author Response
Dear reviewer 1, thanks a lot for revieweing our paper again.
We corrected all the topographical remarks.
Table 3: Kortisol should be Cortisol
We changed this.
Line 139: conrol should be control; please check document on this word. It is written more often as conrol.
We changed this.
Revierwer 2
Dear reviewer 2, thanks a lot for revieweing our paper again.
The term "caecal meteorism" has no meaning in English. Perhaps the authors mean cecal torsion?
No, what we mean is caecum gassing. We changed this.
We have a native english speaker, who translated the manuscript (Philip Saunders, LL.B., B.Ed., FRSA, Language Support Services, E-mail: saundersproof@hotmail.com, Web: www.saundersproof.de).
Reviewer 2 Report
The term "caecal meteorism" has no meaning in English. Perhaps the authors mean cecal torsion?
The manuscript is acceptable now as written, however there is still a lot of editing required to get it into proper English.
Author Response
Dear reviewer 2, thanks a lot for revieweing our paper again.
Revierwer 2
The term "caecal meteorism" has no meaning in English. Perhaps the authors mean cecal torsion?
No, what we mean is caecum gassing. We changed this.
We have a native english speaker, who translated the manuscript (Philip Saunders, LL.B., B.Ed., FRSA, Language Support Services, E-mail: saundersproof@hotmail.com, Web: www.saundersproof.de).